# Mercury-Induced Phytotoxicity and Responses in Upland Cotton (*Gossypium hirsutum* L.) Seedlings

**DOI:** 10.3390/plants10081494

**Published:** 2021-07-21

**Authors:** Lei Mei, Yueyi Zhu, Xianwen Zhang, Xiujuan Zhou, Zhentao Zhong, Huazu Li, Yingjun Li, Xiaohu Li, Muhammad Khan Daud, Jinhong Chen, Shuijin Zhu

**Affiliations:** 1Institution of Crop Science, Zhejiang University, Hangzhou 310058, China; 21916018@zju.edu.cn (Y.Z.); 21816024@zju.edu.cn (X.Z.); 21816016@zju.edu.cn (Z.Z.); 12016016@zju.edu.cn (H.L.); jinhongchen@zju.edu.cn (J.C.); 2Enshi Tujia & Miao Autonomous Prefecture Academy of Agricultural Sciences, Enshi 445000, China; 3The Agricultural Experiment Station, Zhejiang University, Hangzhou 310058, China; bestzxw@zju.edu.cn; 4College of Plant Science and Technology, Huazhong Agricultural University, Wuhan 430070, China; 5Department of Biotechnology and Genetic Engineering, Kohat University of Science and Technology, Kohat 26000, Pakistan; mkdaud@kust.edu.pk

**Keywords:** upland cotton (*Gossypium hirsutum* L.), mercury stress, phytotoxicity, physiological and biochemical response

## Abstract

Cotton is a potential and excellent candidate to balance both agricultural production and remediation of mercury-contained soil, as its main production fiber hardly involves into food chains. However, in cotton, there is known rarely about the tolerance and response to mercury (*Hg*) environments. In this study, the biochemical and physiological damages, in response to *Hg* concentrations (0, 1, 10, 50 and 100 µM), were investigated in upland cotton seedlings. The results on germination of cottonseeds indicated the germination rates were suppressed by high *Hg* levels, as the decrease of percentage was more than 10% at 1000 µM *Hg*. Shoots and roots’ growth were significantly inhibited over 10 µM *Hg*. The inhibitor rates (*IR*) in fresh weight were close in values between shoots and roots, whereas those in dry weight the root growth were more obviously influenced by *Hg*. In comparison of organs, the growth inhibition ranked as root > leaf > stem. The declining of translocation factor (*TF*) opposed the *Hg* level as even low to 0.05 at 50 µM *Hg*. The assimilation in terms of photosynthesis, of cotton plants, was affected negatively by *Hg*, as evidenced from the performances on pigments (*chlorophyll a* and *b*) and gas exchange (Intercellular *CO_2_* concentration (*Ci*), *CO_2_* assimilation rate (*Pn*) and stomatal conductance (*Gs*)). Sick phenotypes on leaf surface included small white zone, shrinking and necrosis. Membrane lipid peroxidation and leakage were *Hg* dose-dependent as indicated by malondialdehyde (*MDA*) content and relative conductivity (*RC*) values in leaves and roots. More than 10 µM *Hg* damaged antioxidant enzyme system in both leaves and roots (*p* < 0.05). Concludingly, 10 µM *Hg* post negative consequences to upland cotton plants in growth, physiology and biochemistry, whereas high phytotoxicity and damage appeared at more than 50 µM *Hg* concentration.

## 1. Introduction

Heavy metal (*HM*) pollution, as a global problem affecting terrestrial and aquatic environments, also posts a potential threat to human health via the food chains. Meanwhile, excessive *HM* concentrations in soils resulted in yield reduction and poor quality of crop products [1]. Mercury (*Hg*) is one of the most toxic *HM* because of its easy bioaccumulation in living bodies [2]. *Hg* may enter agricultural soil through various anthropogenic activities including fertilizers, pesticides, sludge, lime and manure. It exists in nature in forms like elemental mercury, ionic mercury, methyl mercury, mercury sulfide and mercury hydroxide. In all sorts of mercury, ionic mercury is the predominant toxic form [3,4]. In plant, the toxic action of *Hg* is first done on the roots, which take up *Hg* directly in an efficient manner [5]. While entering the root cells, *Hg* may bind to the water channel proteins of root cells and thus causes physical obstruction to water flowing, consequently affecting the transpiration in plants [6,7]. Furthermore, it has also been reported that *Hg* suppresses photosynthesis, chlorophyll synthesis, as well as uptake and transport of nutrients. *Hg* is considered to inhabit the activity of the *NADPH*: protochlorophyllide oxidoreductase (*POR*), which plays key roles in photosynthesis, consequently affects plant growth involving biomass [8]. Sahu evaluated the oxidative damages response to *Hg* concentrations in wheat [9], representing lower and higher *Hg* concentration induced and repressed antioxidant enzymes activities separately, which supported the opinion that *Hg* can boost the formation of reactive oxygen species (*ROS*) and consequently pose more serious oxidative stress on plant cell.

Cottons (*Gossypiums*) are the most important source of natural fiber as well as key sources of edible oil in the world. Due to its hard wooden nature, cotton presents relatively multiple resistances to abiotic stress [10]. Previous works regarding *HM* stresses on cotton revealed varied behavior of cotton cultivars towards various toxic *HM,* like cadmium, chromium, copper and lead. Under sorts of *HM* with concentrations, there appeared anomalies in cotton plants. The performance could be detected, such as a reduction in germination at higher levels of metals, a decline in the biometric parameters like root and shoot length, or fresh and dry biomass, as well as altering on contents of biochemical indictors [11,12,13,14,15]. Regarding the cotton behavior towards *Hg*, no such study has so far been conducted. Keeping in view of the global usage of *Hg* in various industries, the present experiment was designed with the aim to investigate the toxic effects of *Hg* on upland cotton by studying various physiological and biochemical parameters.

## 2. Results

### 2.1. Effects of Hg on Cotton Seeds Germination

In the present study, a *Hg* concentration ranging from 0 to 10,000 µM was employed in order to test the performances on cottonseed germination. Elongation of seed shoot and color diversity, paralleling with the percentage of germination, were studied. Changes in morphology and percentage of seed germination were obvious and presented as Figure 1 and Figure 2. Overall, higher *Hg* concentrations treatment depressed seed shoot elongation, darkened the seeds surface and reduced percentage of germination, comparing with low levels. In details, 1 and 10 µM *Hg* did not affect seed shoot elongation, whereas significances could be observed in the doses 100, 1000 and 10,000 µM. The seed surface color was white in both the control and treatment of 100 µM *Hg*. However, in the groups of 1000 and 10,000 µM, the seed surface turned brown, which was considered as sick. The percentage of germination was tested within a course of 72 h. As revealed in Figure 2, there were no significant variations (*p* < 0.05) in germination percentage from the control to 100 µM, and they maintained percentages above 90% for 72 h. However, the germination percentage decreased starting at 1000 µM *Hg*, and dropped to 80%~90%. At 10,000 µM *Hg*, the germination percentage decreased sharply to less than 10%.

### 2.2. Growth and Hg Distribution

The weight of cotton seedlings declined after *HgCl_2_* treatment for a week. The reduction in both shoots and roots in fresh and dry weight was dose-dependent (Table 1). In shoots, there was no significant difference in fresh weight between the control and 1 µM *Hg* (*p* < 0.05). At 10 µM *Hg*, the mean value of shoots, in terms of fresh weight, was 65.55 g, as the inhibitory rate (*IR*) was 18.8%. When the concentration was over 50 µM, the *IR* seemed to approach the summit. In details, the values were 36.26% and 37.32% at 50 and 100 µM *Hg,* separately, which were quite close each other. Regarding the dry weight in shoots, it can be affected by elevated *Hg* concentrations and exhibited analogous trends with that in fresh weight, even though there was no statistical difference at 10 µM *Hg*. Correspondingly, the growth inhibition of both fresh and dry weight in roots was in a dose-dependent manner with increased *Hg* level. What is more, the significant *IR* change alternativeness (comparing with the controls) on fresh and dry weight of roots began at 10 µM *Hg*, and the IR value was over 32.10 at 50 and 100 µM *Hg*. Overall, the cotton seedlings treated by 10 µM *Hg* for a week will show significant growth inhibition and inhibitory upper limit appeared at approximately 50 µM *Hg*. Comparing to shoots as well as fresh weight separately, the growth of relevant roots and dry weight was more easily suppressed by *Hg* toxicity.

*Hg* was accumulated in all organs involved in leaves, stems and roots after the treatments (Table 2). The roots concentrated most *Hg*, followed by leaves and stems. The concentration index ranged from 1 to 58.89, 1 to 7.22 and 1 to 4.02 in those 3 sorts of organs, respectively and orderly. As a whole on means data, *Hg* enrichment in roots was 8- and 16-fold that in leaves and stems separately. With the increased *Hg* concentrations, the translocation factor (*TF*) in cotton seedlings decreased. At 50 and 100 µM *Hg*, the *TF* value declined to 0.05 and 0.06, respectively, which represented the limit of *Hg* translocation.

### 2.3. Effects of Hg on Cotton Assimilation

The growth of treated plants was inhibited under *Hg* stress, which may indicate the decline on assimilations regarding photosynthesis. The phenotypes on leaf surface, as well as pigments and gas exchange involved in photosynthesis, were studied. Comparingly, much severe sick phenotypes appeared on leaves under higher *Hg* concentration (Figure 3). At 10 µM *Hg*, small white zone emerged on the leaf upper surface, whereas obvious necrosis and shrinking happened at 50 and 100 µM *Hg*. Obviously, the extent of leaf health inversely synchronized to the *IR* depending on *Hg* levels.

The average amounts of photosynthesis pigments, *chlorophyll a* and *b*, showed a similar decreased trend exposure to more severe *Hg* pollution (Figure 4). *Chlorophyll a* decreased to 93.2%, 74.6%, 51.5% and 36.9% at 1, 10, 50 and 100 μM *Hg* concentration, respectively, in comparison with the control. Meanwhile, 98.8%, 74.0%, 53.5% and 47.2% declines were found in *chlorophyll b* as compared with the control. The total *chlorophyll* content, comprised of *chlorophyll a* plus *b*, exhibited the declining trends. In comparison to the control, the total *chlorophyll* reduced to 94.1%, 74.5%, 51.8% and 38.5% at *Hg* treatments of 1, 10, 50 and 100 µM, respectively.

Determination on leaf gas exchange parameters must help to track photosynthesis efficiency influenced by *Hg* stress (Figure 5). Compared with the control value 283.54 µmol·mol^−1^, *Ci* was increased to 290.34, 340.15, 380.31 and 395.21 µmol·mol^−1^ at 1, 10, 50 and 100 µM *Hg* treatments, correspondingly. A significant reduction in *Gs* was noted at 50 and 100 µM *Hg* treatments, and those were decreased to 67.11% and 53.95% in comparison to the control. On the contrary, *Gs* value in 1 and 10 μM *Hg* treatments did not present a significant increase compared to the control (Figure 5B). Concerning *Pn*, there was no significant difference between 1 µM *Hg* and the control. However, profound value gaps on *Pn* were shown at 10, 50 and 100 μM *Hg*, as reduced to 49.75%, 19.54% and 12.59%, separately.

### 2.4. Membrane Lipid Peroxidation and Leakage

Accumulation of malondialdehyde (*MDA*), as a key indicator of membrane lipid peroxidation, is commonly employed to evaluate cell injury under stresses [16]. *MDA* has been induced in both leaves and roots with the raising of treated *Hg* concentrations. As a whole, *MDA* accumulations in leaves were estimated 6-fold on average as much as that in roots (Figure 6). In leaves, amounts of *MDA* were increased apparently to 1.30, 1.35 and 1.46 times at 10, 50 and 100 µM *Hg* treatments, respectively, in comparison with the control, whereas those increased to 1.54, 1.80 and 2.28 times in roots, correspondingly.

Effects of variable *Hg* stresses on electrolyte leakage (*EL*), in both leaves and roots, were investigated via relative conductivity (*RC*) (Table 3). *Hg* stress induced increase of *EL* significantly. In leaves, the *RC* value in the control was 15.54%, and that was 15.85%, 30.30%, 53.95% and 78.36% at 1, 10, 50 and 100 μM *Hg*, respectively. In roots, correspondingly, *RC* was 35.22%, 38.64%, 52.10% and 54.01% at *Hg* treatments, which were significantly much more than that in the control as value 20.21%.

### 2.5. Antioxidant Enzyme Systems Response

Antioxidant enzymes were analyzed in both roots and leaves, in order to study their response in scavenging of *ROS* during the course of external application of *Hg*. *SOD* activities, both in leaves and roots, were inhibited gradually with the rising of *Hg* concentrations. Wholly, the *SOD* activities of leaves were higher than that of roots (Figure 7A and Figure 8A). In leaves and comparing with the control, there was no significant decrease at 1 and 10 µM *Hg*. However, the great decrease of values appeared at both 50 and 100 µM. However, *SOD* activities of treated plants went down to 84.5%, 79.3%, 63.8% and 59.3%, correspondingly, relative to the untreated control in roots. Concerning *POD*, similar trends were shown in both leaves and roots as *SOD* activity. *POD* activities were apparently lower in leaves than those in roots wholly. In comparison to the control value 8.836 mM g^−1^ FW min^−1^, POD activities were decreased to 8.031, 6.324, 4.324 and 3.986 mM g^−1^ FW min^−1^ at 1, 10, 50 and 100 µM *Hg* in leaves, respectively (Figure 7B). Correspondingly, in roots, those values were 26.350, 23.240, 19.340, 13.250 and 11.648 mM g^−1^ FW min^−1^ from 0 to 100 µM *Hg* treatment (Figure 8B). As regards *CAT,* the activities were suppressed profoundly by high-level *Hg* treatments in both leaves and roots (Figure 7C and Figure 8C). *CAT* activities in leaves decreased to 50.93% and 45.99% at 50 and 100 µM *Hg* as compared with control, and those decreased to 49.37% and 43.04% in roots, separately. The activity was 0.082 mMg^−1^ FW min^−1^ at 1 µM *Hg* in roots, which was higher than the control 0.079 mMg^−1^ FW min^−1^, although there was no statistical difference. As shown in Figure 7D and Figure 8D, *APX* activities were inhibited significantly at 10, 50 and 100 µM *Hg*, in both leaves and roots. Furthermore, the activity bottom appeared at concentration of 100 µM *Hg*, as 1.135 and 0.532 mM g^−1^ FW min^−1^ in leaves and roots, separately. However, there was no significance between the control and 1 µM level.

## 3. Discussion

Mercury (*Hg*), as a highly toxic nonessential element and its dispersion in the environment, is considered as a serious environmental problem for its persistent character [17]. Unlike most metals that function as nutrients, *Hg* has no known physiological action, so it is not metabolized by most organisms [2]. Knowledges in terms of the toxic effects of *Hg* in plants may be conducive to the general understanding on the primary toxicity mechanism and the tolerant characters in living organisms such as upland cotton.

### 3.1. Seeds Versus Seedling Plants Exhibited Higher Tolerance to Hg Toxicity

Seed germination and seedling growth are convenient and simple end points to be employed for evaluating the toxicity of pollutants to higher plants at the early stages, which are associated with biomass directly [18]. In the present study, a wide range of *Hg* concentration treatments from 0 to 10,000 µM was chosen for detection of inhibition on germination. The results on germination indicated that cottonseed was able to bear extra-high *Hg* concentration stress. Seed *GR* maintained over 80%, although the *Hg* concentration was as much as 1000 µM. Significant differences were easily detected on the fresh or dry weight of both leaves and roots separately even in the 10 µM treated plants, compared to the relevant controls. As a consequence, cotton seeds exhibited much more tolerance to *Hg* stress, in comparison with the seedling plants. In addition, 50 µM *Hg* stress never affected the seed germination of *Quercus ilex*, but the root fresh weight and longest root length were easily depressed even by 5 μM *Hg* [19]. Ling reported that seed germination was not repressed at 100 µM *Hg* concentration in four kinds of vegetables including cabbage (*Brassica rapa*), cole (*B. napus*), head cabbage (*B. oleracea*) and spinach (*Spinacia oleracea*) [20]. Additionally, the wheat seed *GR* decreased by approximately merely 10% under the *Hg* concentration as high as 1000 μM [21]. In present experiments in seedlings, their growth seemed to be inhibited more than 10 µM *Hg*. It is obvious that *Hg^2+^* are likely to be aborted by cotton root and subsequently transported to the whole plant. Our results are consistent with documented species. It is clear that either woody or herbal plants showed relatively more tolerance and sensitivity to *Hg* environment in seeds and plants, respectively. Seeds, as the key energy bank to organism survival, store amounts of nutrition and process stronger vigor comparing with other organs, probably conferring the ability to face the change on the environment involving *HM*. Roots are the primary organ to contact the *Hg*, which could be distributed to the overall plant via the phloem. Phytotoxicity of *Hg* resulted from loss of function on targeted aquaporin [22,23]; thus, this protein family in cotton is sensitive to *Hg*. In wheat, significant phytotoxicity was at above 10 μM *Hg* concentration, as the phytotoxicity appeared in upland cotton. Thus, cotton seedling exhibited some native tolerance to *Hg* toxicity, whereas it seemed not too apparent the advantages to other main field crops [9]. Further and comprehensive studies need to be conducted in order to reveal the spatial and temporal responses of Hg on cotton plants, and those would enlighten the roads to remediate *Hg*-containing soil via cotton.

### 3.2. Damage of Photosynthesis Resulting from Hg Toxicity Accounted for the Sick Growth

Plant growth is the function in terms of cell wall extensibility, osmotic potential, water conductivity, and threshold turgor, among other factors. Growth does not occur when these agents are insufficient [24]. Decrease in seedling growth involving biomass has been well documented in plants under *Hg* stress. This might be attributed to the lack of essential elements such as *Fe*, *S* and *Zn*, and decreasing photosynthesis as *Hg* poison [8]. As inhibition on growth in response to *Hg* toxicity in upland cotton, a similar response has also been detected in soybean (*Glycine max*), radish (*Raphanussativus*), tomato (*Solanum lycopersicum*) and other plants [25,26]. *Hg* also damaged photosynthetic pigments, which represented a declining of contents of chlorophyll. It was reported that HM depresses chlorophyll formation via interfering with protochlorophyllide production [27]. HM might be also replaced by the Mg from chlorophyll molecules in green plants [28], and thereby chlorophyll pigments reduce the photosynthetic efficiency undoubtedly.

The leaf gas exchange study indicated that photosynthetic parameters, referring to *Gs*, *Pn* and *Ci*, were associated with the inhibition of growth due to *Hg* stress. Those were reported in plants such as white lupin (*Lupinus albus* L.), rice (*Oryza sativa* L.) and Arabidopsis (*Arabidopsis thaliana*) [29,30]. In addition, these agree with our results that a rise in intercellular *CO_2_* concentration and a decline in both stomatal conductance and *CO_2_* assimilation rate are due to increasing *Hg* levels. Thus, reduction of biomass under *Hg* stress may be the consequence of reduction in photosynthesis pigments and following blocking of photosynthesis. Declined *Pn* may result from the inhibition of relevant reaction steps in the *Benson–Calvin* cycle and rubisco activity influenced by these contaminants [31]. Moreover, a decrease in Pn could be due to stomatal closing.

### 3.3. Decrease on inTegrity of Membrane-Symbolled Cellular Injure Resulting from Hg Toxicity

Being analogous to other abiotic stress, HMs damage membrane is largely mediated via membrane lipid peroxidation, leading to *EL* [16]. It is reported that an increase in *MDA* contents in response to *Hg* stress has been detected, both in leaves and roots, from *Medicageo sativa* and other plants [32]. Significant accumulations of *MDA* content were observed, both in leaves and roots, from treatments which were more than 10 µM *Hg* concentration, in upland cotton plants. These results imply that the membranes are injured in consequence of reactive oxygen species, as cotton seedlings are contaminated by high *Hg* levels. Rising of percentage in regard to electrolytic leakage has been observed both in the leaves and roots of upland cotton seedling treated with higher than 10 µM *Hg* concentration as well. It is concluded that root, rather than leaf, is more sensitive to *Hg* stress, resulting from the fact that significant variances were detected in 1 µM *Hg* level in roots but not in leaves, as compared with the controls, respectively, which is in agreement with the result of *SOD*. The sensitivity to *Hg* stress was relatively serious in roots as compared to leaves, which may be attributed to *Hg* accumulation in roots being faster as they are exposed to *Hg* iron directly. Lipid peroxidation resulting from *Hg* stress has been reported in many plants, and Mishra have found an increase both of *MDA* and electrical conductivity in *Oryza sativa* by treated with *Hg* concentration, which is in line with our results in upland cotton [33].

### 3.4. Hg Disturbed Cellular Redox Equilibrium Involving Depression on Activities of Antioxidant Enzyme System

As a transition element, oxidative stress can be induced by *Hg* in consequence of excess production of reactive oxygen species (*ROS*) in plants, leading to alternative antioxidant enzymes activities, lipid peroxidation and ion leakage [34]. Plants, with a set of potential mechanisms, might be associated with metal detoxification at the cellular level, and tolerance in plant is shown in HMs. The generation of ROS induced by *Hg* triggers influences activation of components in the antioxidative defense system in plants, as it consists of both the enzymatic system, which includes *SOD*, *POD*, *APX*, *CAT* and others, and the non-enzymatic system, containing some low-molecular-weight antioxidants such as glutathione and ascorbic acid, etc. Overall, exposure to higher concentrations of *Hg* concerning 50 and 100 µM altered the activities of all four sorts of antioxidant enzymes, both in leaves and root, in which *Hg* ions at low level (1µM) did not lead to significant alternation, although, as compared with the controls, respectively. In conclusion, relevant activities of antioxidant enzymes are suppressed just under higher *Hg* concentration, rather than lower *Hg* treatment. The enzymatic antioxidant *SOD*, as the first enzyme in the detoxifying process, is considered an essential component among antioxidant defense systems in plants [35,36]. Root is more sensitive to *Hg* than leaf, resulting from the fact that more significant variances, in terms of *SOD*, were detected in roots rather than in leaves in lower *Hg* levels, including 1 and 10 μM. The changes in activity of *POD* have been observed under serious *Hg* levels. The trend in leaves is in line with that in roots, thus it is hypothesized that *POD* activity might be influenced in leaf and root approximately. Regarding *CAT*, the activity was inhabited by *Hg* levels both in leaves and roots overall, while that was slightly enhanced in 1 µM *Hg* concentration in roots. It shows that *CAT* activity could be triggered by the lower concentration in roots to some extent. It has been well documented that some sorts of heave metal with low concentration stimulated activities of antioxidant enzymes, differing from the sort of the enzymes, *Hg* concentration levels and plants species [9,37,38,39]. In response to *Hg* stress, activity of APX exhibited parallel trends with that of *POD,* both in leaves and roots. Whether direct linkage exists between *POD* and *APX* in response to *Hg* stress in upland cotton should be further studied.

## 4. Materials and Methods

### 4.1. Assay on Seed Germination

Matured uniform-sized seeds of upland cotton standard species *Texas-Marker-1* (*TM-1*) were decoated and sterilized by 70% ethyl alcohol for 5 min, and then treated by 0.1% (*w*/*v*) *HgCl_2_* for 5 min. They were thoroughly washed with distilled water more than 4 times. Aiming at examining *Hg* tolerance extent in upland cotton, a wider range of concentration gradient was employed in germination experiments via 0, 1, 10, 100, 1000 and 10,000 µM and in the form of *HgCl_2_*. Every 20 seeds were allocated into flasks with different *Hg* salt concentration. The flasks were placed in dark conditions in an orbital shaker (Thermo Fisher Scientific, Waltham, MA, USA) at 120 rpm at 30 °C. The percentages of seeds germination were counted every 24 h, 3 times in total.

The germination rate was calculated as the formula:(1)$$GR\;(\%)=\frac{\mathrm{means}\;\mathrm{of}\;\mathrm{germmination}\;\mathrm{seeds}}{\mathrm{means}\;\mathrm{of}\;\mathrm{total}\;\mathrm{seeds}\;}\times 100\;$$

### 4.2. Investigation on Growth and Hg Distribution

The germinating seeds were kept in a dark room at 28 ± 2 °C, 65% relative humidity, for 3 days, and then the baby seedlings were transferred to growth chambers with hydroponic solution condition (1/8 *Murashige and Skoog* concentration), under light/dark: 16 h/8 h, before emergence of true leaf. The seedlings were permitted to grow for 2 weeks in the hydroponic media and the solution were refreshed every 2 days. The 15-day-old seedlings were treated by *HgCl_2_* for a week. In this experiment, the concentrations treated were chosen as 0, 1, 10, 50, 100 µM according to the germination performance responding to the previous wider dose border. Fresh and dry roots, stems and leaves were weighed from 3 individuals as a replicate, a total of 9 seedlings were used for each concentration per treatment. The inhibitory extent in seedling growth evaluated by inhibitory rate (*IR*, referring the following formula (2). To determine *Hg* content, approximate 0.2 g dry samples were proceeded by nitric acid digestion for upcoming *Hg* element quantification by Atomic Fluorescence Spectrometer referring to [40]. *Hg* concentrated in tissues and translocation calculated by formulas (3) and (4).
(2)$$\mathrm{Inhibitory}\;\mathrm{Rate}\;(IR)=\frac{\mathrm{Weight}\;\left(\mathrm{Control}\right)\;–\;\mathrm{Weight}\;\left(\mathrm{Treated}\right)}{\mathrm{Weight}\;\left(\mathrm{Control}\right)\;}$$
(3)$$\mathrm{Concentration}\;\mathrm{Index}\;(CI)=\frac{\mathrm{Hg}\;\mathrm{contents}\;\mathrm{in}\;\mathrm{treated}\;\mathrm{plant}\;\mathrm{part}}{\mathrm{Hg}\;\mathrm{contents}\;\mathrm{in}\;\mathrm{the}\;\mathrm{control}\;}\;$$
(4)$$\mathrm{Translocation}\mathrm{Factor}\;(\mathrm{TF})=\frac{Hg\;contents\;in\;shooter}{Hg\;contents\;in\;root\;}$$

### 4.3. Measures of Pigments and Gas Exchanges in Leaves

Measurement of chlorophyll pigments was conducted according to the protocol of Porra [41]. To determine chlorophyll *a* and chlorophyll *b* and total chlorophyll contents, reaction solution was made using pure acetone, pure ethanol and distilled water were mixed in the ratio of 4.5: 4.5:1. The 0.5 g small leaf pieces were taken in a 25 mL tube and placed in dark until the color of the sample convert into white. Values of *OD_663_* and *OD_645_* were read by UV-spectral. Chlorophyll contents were determined using the following formulae:*Chl. (a)* (mg/L) = 9.78OD_663_ − 0.99*OD_645_*(5)
*Chl. (b)* (mg/L) = 21.43*OD*_645_ − 4.65*OD*_663_(6)
*Chl. (t)* = *Chl. (a)* + *Chl. (b)*(7)
where *Chl. (a)*, *Chl. (b)* and *Chl. (t)* are abbreviated from chlorophyll *a*, chlorophyll *b* and total chlorophyll respectively.

Gas exchange parameters, including *CO_2_* assimilation rate (*Pn*), stomatal conductance (*Gs*) and the intercellular *CO_2_* concentration (*Ci*), were determined by an infrared gas analyzers portable photosynthesis system (LI-COR 6400, Lincoln, NE, USA), using third fully expanded leaves. The measurement was conducted during 10:00–11:00 a.m., maintaining the air temperature, air relative humidity, photosynthetic photon flax density (PPFD), and *CO_2_* concentration at 25 °C, 75–85%, 1000 µMol m^−2^ s^−1^ and 400 µMol m^−2^ s^−1^, respectively.

### 4.4. Assay on Malondialdehyde (MDA) Content and Electrolyte Leakage (EL)

Melondialdehyde (*MDA*) contents were determined according to the protocol describe by Zhou [42]. Briefly, 0.5 g root or leaf samples was homogenized in tube with 10 mL 0.5% *TBA*, followed by heating the homogenate at 95 °C for 30 min and then cooled on ice immediately. After that, it was centrifuged at 5000 rpm for 8 min and the supernatant was taken for measure at 532 nm and 600 nm. The determination of *MDA* was followed as the formula:(8)$$MDA\;(\mathrm{nmol}/g\;\mathrm{FW})=\frac{\left(\mathrm{OD}532-\mathrm{OD}600\right)\times \;A\;\times \;V}{a\;\times \;E\;\times \;W}$$
where A = Volume of total reaction solution and enzyme extract, V = Total volume of PBS used, a = Volume of the enzyme extract used, W = Fresh weight of the sample and E = Constant for *MDA* (1.55 × 10^−1^).

Leaf and root samples were divided into small pieces of about 5 mm in size respectively. Electrolyte leakage (*EL*) was estimated referring to *Dionisio-Sese* and *Tobita* (1998) and the relative conductivity (*RC*) was calculated via the following formula,
(9)$$RC\;(\%)=\frac{\mathrm{Primary}\;\mathrm{value}\;\mathrm{of}\;\mathrm{electric}\;\mathrm{conductivity}}{\mathrm{Value}\;\mathrm{of}\;\mathrm{electric}\;\mathrm{conductivity}\;\mathrm{via}\;\mathrm{killing}\;\mathrm{out}\;}\times 100\%$$

### 4.5. Determination of Antioxidant Enzyme Activities in Cotton Seedlings

For the determination of antioxidant enzymes’ activities, 0.5 g fresh samples of roots or leaves were homogenized with 3 mL potassium phosphate buffer (*PBS*, 50 mM and *pH* 7.8) in prechilled mortar and pestle, and then finally the volume of the homogenate was made to be 8 mL with further addition of *BPS*. The homogenate was centrifuged for 20 min at 12,000 rpm at 4 °C, and the supernatant was preserved for determination of antioxidant enzymes’ activities. The assays for the determination of various antioxidants were performed according to the established protocols described by Daud [11].

### 4.6. Statistical Analyses Plata

In this study, the data were statistically analyzed based on one-way analysis of variance (*ANOVA*) by software SAS (Version 9) (SAS Institute INC, Raleigh, NC, USA). All test samples contained 3 replicates. Means were separated at 5% level of significance by least significant difference (*LSD*) test.

## 5. Conclusions

In conclusion, cotton seeds versus plants had a higher tolerance to *Hg* stresses. In view of the data on means, involving *GR*, growth parameters, pigments, gas exchanges, *RC* and the actives of antioxidant enzymes, significant alterations appeared within cotton seeds and seedlings, at 1000 and 10 µM *Hg*, respectively. In plants, relatively severe phytotoxicity and damage in response to *Hg* started above the 50 µM level. *Hg* distribution in cotton plants affected by *Hg* levels and root detained more *Hg* ion compared to shoots. As a whole, in cotton plants, the ability to concentrate and transport *Hg* ion seemed to close the limitation at over 50 µM concentrations. Overall, cotton seedlings showed some natural tolerance to *Hg* toxicity, whereas there was no apparent advantage to other field crops.

## Figures and Tables

**Figure 1 plants-10-01494-f001:**
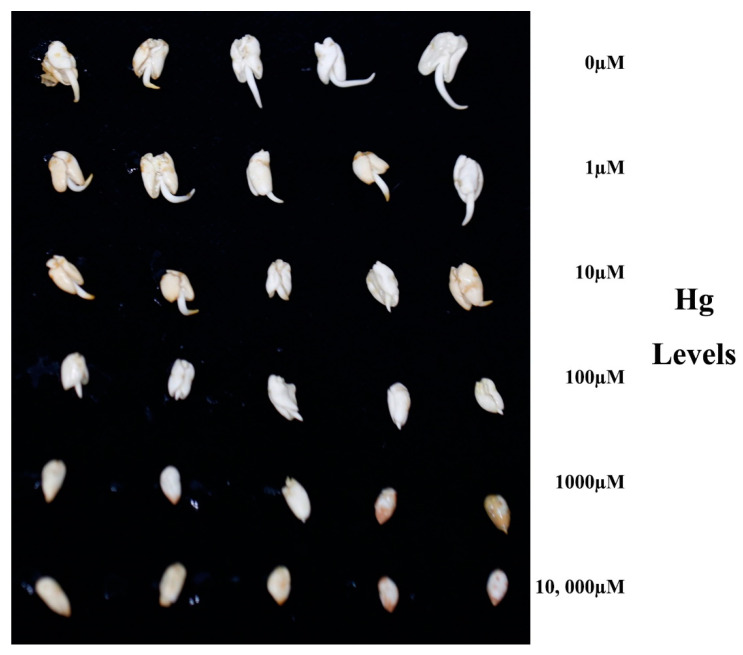
Morphology on germination of decoated seeds *TM-1* under *Hg* treatments. Decoated seeds exposed to *Hg^2+^* solution with gradient 0, 1, 10, 100, 1000 and 10,000 µM, which were presented from upper to lower rows respectively.

**Figure 2 plants-10-01494-f002:**
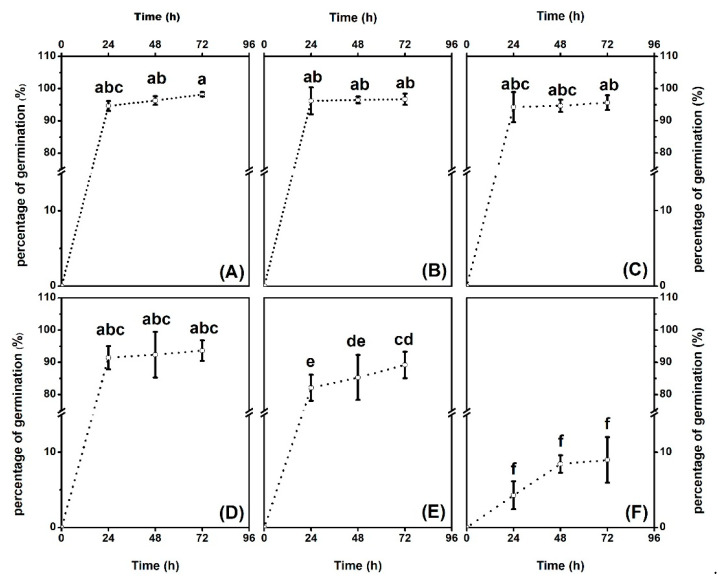
Germination rate of decoated cotton seeds under *Hg* concentrations during a course of 72 h. (**A**–**F**): De-coated cotton seeds exposed to *Hg^2+^* solution with concentration at 0, 1, 10, 100, 1000 and 10,000 μM, respectively, and the germination experiment was conducted under shaking condition at 130 rpm and 28 °C in dark. The percentages of germination were determined at 0, 24, 48 and 72 h. Error bars represent *SD* value (*n* = 3). Different letters indicate significant differences (*p* < 0.05).

**Figure 3 plants-10-01494-f003:**
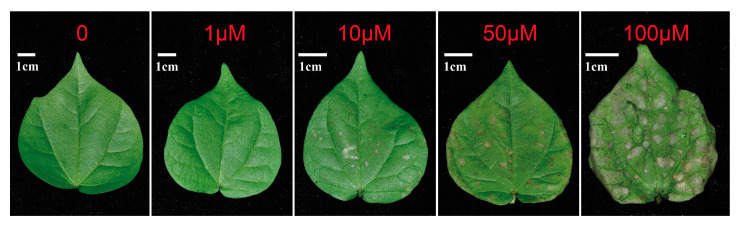
Morphological leaves responding to *Hg* levels at 0, 1, 10, 50 and 100 µM. The scale bars denote 1 cm.

**Figure 4 plants-10-01494-f004:**
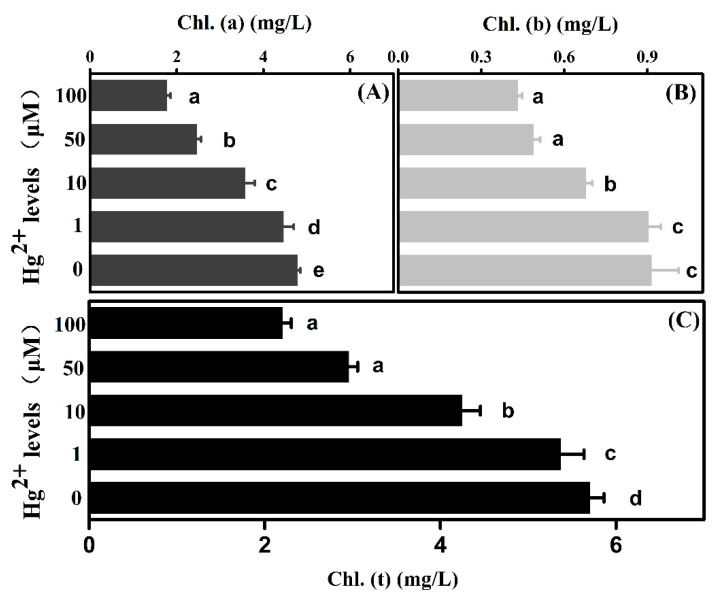
Pigment content in leaves under stress from *Hg* levels. (**A**–**C**): *chlorophyll a*, *chlorophyll b* and total *chlorophyll* contents, as abbreviated to *Chl. (a)*, *Chl. (b)* and *Chl. (t)*, are shown, respectively. Error bars represent *SD* value (*n* = 3). Different letters indicate significant differences (*p* < 0.05).

**Figure 5 plants-10-01494-f005:**
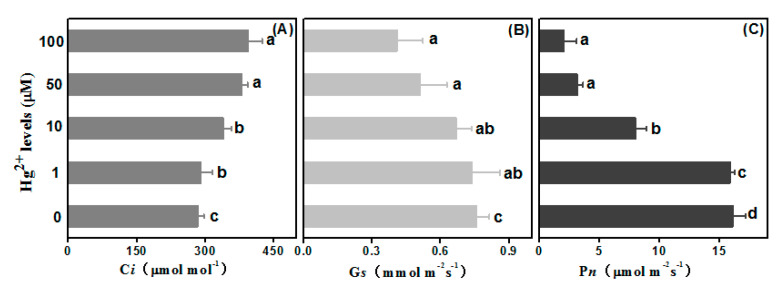
Gas exchange in leaves under stress from *Hg* levels. The column (**A**–**C**) are exhibited as the intercellular *CO_2_* concentration (*Ci*), stomatal conductance (*Gs*) and *CO_2_* assimilation rate (*Pn*), respectively. Error bars represent *SD* value (*n* = 3). Different letters indicate significant differences (*p* < 0.05).

**Figure 6 plants-10-01494-f006:**
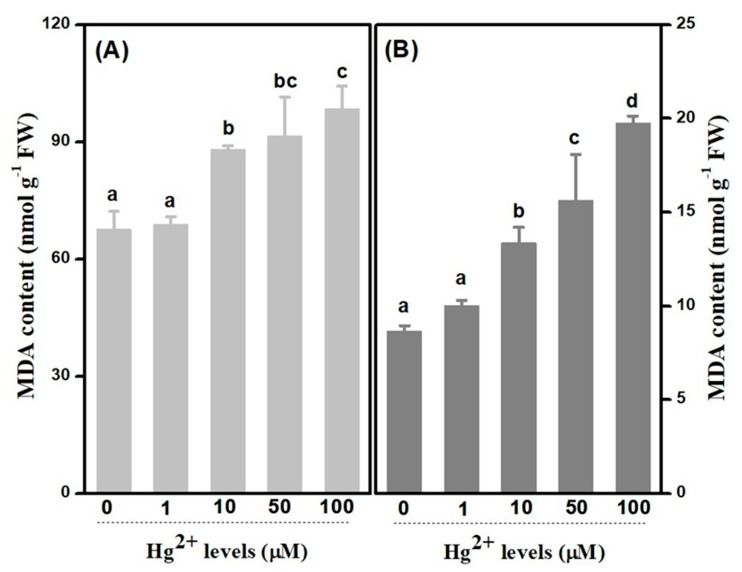
*MDA* content under *Hg* levels in leaves and roots. (**A**): *MDA* content under *Hg^2+^* levels in leaves; (**B**): *MDA* content under *Hg^2+^* levels in roots. Error bars represent *SD* value (*n* = 3) and different letters indicate significant differences (*p* < 0.05).

**Figure 7 plants-10-01494-f007:**
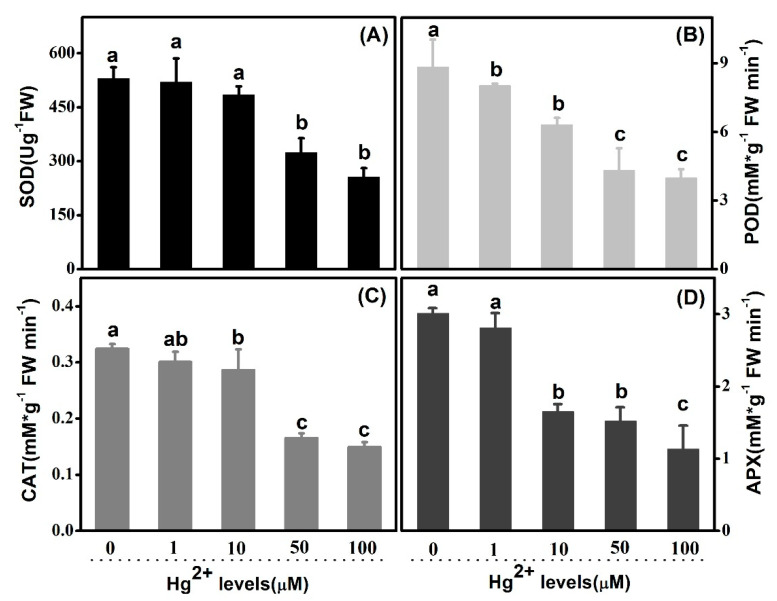
Effects of *Hg* on *SOD*, *POD*, *CAT* and *APX* activity in functional leaves. (**A**–**D**): *SOD*, *POD*, *CAT* and *APX* activity is shown successively and respectively. Error bars represent *SD* value (*n* = 3). Different letters indicate significant differences (*p* < 0.05).

**Figure 8 plants-10-01494-f008:**
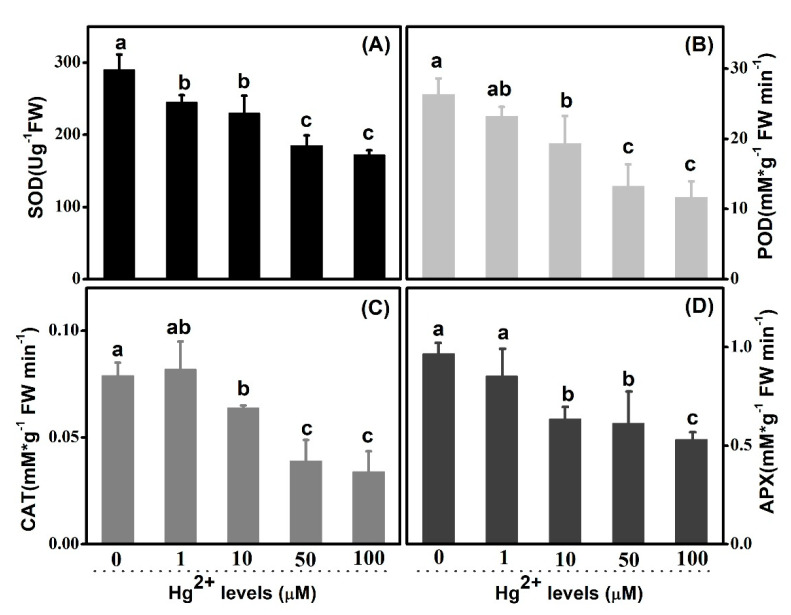
Effects of *Hg* on *SOD*, *POD*, *CAT* and *APX* activity in roots. (**A**–**D**): *SOD*, *POD*, *CAT* and *APX* activity is shown successively and respectively. Error bars represent *SD* value (*n* = 3). Different letters indicate significant differences (*p* < 0.05).

**Table 1 plants-10-01494-t001:** Effects of *Hg*^2+^ levels on growth in shoots and roots.

*Hg^2+^* Treatment (μM/L)	Shoots				Roots			
	Fresh Weight (g)	IR (FW) (%)	Dry Weight (g)	IR (DW) (%)	Fresh Weight (g)	IR (FW) (%)	Dry Weight (g)	IR (DW) (%)
0	80.73 ± 1.22 a	0	6.76 ± 0.69 a	0	21.62 ± 0.06 a	0	1.91 ± 0.28 a	0
1	79.89 ± 1.97 a	1.04	6.54 ± 0.44 a	3.25	20.78 ± 1.03 a	3.89	1.81 ± 0.10 a	5.24
10	65.55 ± 5.37 b	18.8	5.95 ± 0.69 ab	11.98	17.58 ± 1.36 b	18.69	1.51 ± 0.13 b	20.94
50	51.42 ± 3.23 c	36.26	5.42 ± 0.11 b	19.82	14.68 ± 1.90 c	32.10	1.07 ± 0.15 c	43.98
100	50.58 ± 4.14 c	37.32	5.40 ± 0.10 b	20.19	13.91 ± 0.69 c	36.16	0.94 ± 0.07 c	49.21

Error bars represent SD value (*n* = 3) and different letters indicate significant differences (*p* < 0.05).

**Table 2 plants-10-01494-t002:** Mercury distributions in tissues and their concentration and translocation in seedlings.

*Hg^2+^* Treatments (μM)	Hg Concentration (μgg^−1^ *DW*)			*CI* (Concentration Index)			*TF* (Translocation Factor)
	Leaves	Stems	Root	Leaves	Stems	Roots	
0	1.45 ± 0.10 a	1.63 ± 0.28 a	5.16 ± 0.05 a	1.00	1.00	1.00	0.60
1	2.23 ± 0.12 b	2.28 ± 0.19 b	28.60 ± 0.36 b	1.54	1.40	5.54	0.16
10	3.86 ± 0.10 b	2.42 ± 0.01 c	54.81 ± 0.33 c	2.66	1.48	10.62	0.11
50	9.37 ± 0.45 c	4.05 ± 0.06 d	288.23 ± 20.75 d	6.46	2.48	55.86	0.05
100	10.47 ± 0.27 d	6.56 ± 0.01 e	303.89 ± 11.58 d	7.22	4.02	58.89	0.06

Error bars represent *SD* value (*n* = 3) and different letters in same group indicate significant differences (*p* < 0.05). μgg^−1^
*DW* indicates the amount in terms of microgram *Hg* per gram in dry tissue sample.

**Table 3 plants-10-01494-t003:** Effects of *Hg^2+^* levels on relative conductivity (*RC*) in leaves and roots.

*Hg^2+^* Levels (μM)	Relative Conductivity (RC) (%)	
	Leaves	Roots
0	15.54 ± 1.83 a	20.21 ± 3.53 a
1	15.85 ± 2.83 a	35.22 ± 3.59 b
10	30.30 ± 7.81 b	38.64 ± 0.87 b
50	53.95 ± 5.07 c	52.10 ± 7.73 c
100	78.36 ± 8.68 d	54.01 ± 5.97 c

Error bars represent SD value (*n* = 3) and different letters indicate significant differences (*p* < 0.05).

## Data Availability

Not applicable.

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
