# Peer review of "Mercury-Induced Phytotoxicity and Responses in Upland Cotton (Gossypium hirsutum L.) Seedlings"

_plants, 2021, doi:10.3390/plants10081494_

Round 1
Reviewer 1 Report
The paper discusses the problem of the presence of Hg in the environment.
The authors have presented a number of studies describing the effect of concentration on germination, growth and individual parts of plants as well as accumulation of Hg in plant tissues. Aspects of the effect of Hg on chlorophyll levels and the physiological and biochemical responses of Upland Cotton are also discussed.
It is worth comparing these studies with those carried out in the soil system.
However, the work needs to be carefully reviewed and corrected. The work contains errors in spelling formulas of chemical compounds, units, Latin names of plants are written in italics. Moreover, it is not necessary to provide symbols of elements in brackets next to their names, as it is commonly used.
The degree of oxidation of an element should be written as a superscript.
Instead, abbreviations should appear at the first mention of the parameter analysed. It is very important that the abbreviation is used unchanged throughout the manuscript, maintaining the letter size and font type.
e.g., the abbreviation for inhibitory rate is IR or RI.
Explanation of letters (a, b, c, d, e) is missing in Figures 4, 5 6,7 and 7; and in Table 3. There is an error in the numbering of figures. There are two numbers 7.
Please read the sentence in chapter 3.2. lin266 - 268. is S - sulphur or the authors are sure it is a metal?
Please check the abbreviation of phosphate buffer.
In conclusion, the first sentence adds nothing to the paper. I suggest deleting it.
Author Response
Please see the attachment, cheers.

Reviewer 2 Report
The work presents numerous data, however, many inaccuracies of various kinds should be corrected. I have reported my comments and suggestions below. English should be revised
The abstract is acceptable and summarizes the study conducted well. There are a few writing errors that need to be corrected.
Insert the concentrations of Hg investigated in this study
Lane 17: "The" must be written in lower case
Lane 18: “ results on’cottonseeds” correct
The introduction provides the information necessary to contextualize the study conducted. It is written in a very concise manner. In particular, the goals and objectives of the work are hastily described. Therefore, I recommend rewriting the concluding part of the introduction.
The references’ format has not been respected, so it must be corrected throughout the text.
Lanes 37-40: rephrase the sentence in a clearer form
Lane 43: insert a reference supporting the sentence
Results
Lines 74-75: avoid the word laboratory
Lines 82-83: please rephrase the sentence
Table 1: insert in the table the unit for the fresh and dry weight
Table 2: correct the unit of measure for Hg concentration
Line 132: the decrease in the assimilation of what? Try to explain better
Lines 133-134: please rephrase in a clearer form
Lines 135-137: quite obvious
Line 148: “mol-1” put in superscript
Figure 3: inserts a scale in the photos, if possible, to give readers an idea of the leaf size. Looking at it this way it looks like there is no effect on the biomass.
Figure 4 and the discussion on the chlorophyll a and b. Typically, chlorophyll a is present relatively to b in a 3:1 ratio. From the figure, it looks like the opposite. Could it be that the figure has been reversed? This point must be clarified
Figure 5: carbon dioxide must be written with lower case
We have two Figures 7. I suggest calling the one on the activities of antioxidant enzymes at the radical level as Figure 8
Figures 7D are missing the bar errors
Discussion
Lines 228-232 could be eliminated
Line 236: Maybe you mean wide
Lines 237-238: I would talk more about inhibition than seed tolerance
Lines 239-240: This result seems at odds with the picture in Figure 1. Even concentration 1000 seems to inhibit germination almost completely.
Lines 252- 253: verify the concentration
Line 253: correct Hg2+
Lines 281-282: correct CO2 with subscript
Line 286-287: insert a reference in support of your conclusion
Line 325: Please insert a reference
M&M
Line 343: write HgCl2 correctly and all formulas in the materials and methods section
Line 362: In Table 2, the mercury content is referred to dry weight. In this part of Materials and Methods, the authors say they used 0.2 grams of fresh weight. Please clarify this point.
Line 368: in the formula, I don’t think the right word is shooters
Line 400: indicate the weight of root and shoot for EL determination
Lines 402-403: better explain how RC was estimated
CONCLUSION
Lines 417-418 must be deleted
Lines 421-422: Since antioxidant enzymes were not stimulated at the lowest concentrations, the conclusions are not supported. Please explain.
Reviewer 3 Report
The study is interesting but has some major flaws that must be corrected before publication. The main flaw in the manuscript is the writing because many parts of the manuscript are difficult to read. Many words are misspelled (e.g. flax, grown, wild, etc) and the English language and style must be improved. I recommend that the manuscript be reviewed by a native speaker with scientific experience.
Many used references are old. You should change them for newer references
Subindex and superindex are not properly used (e. g. HgCl2, CO2, and units for parameters)
Use abbreviations for Mercury (Hg) and Heavy metal (HM).
Use italics for the scientific names of species
ABSTRACT
L23. Why do you mention the stem? You do not include data about stem in the manuscript
MATERIAL AND METHODS
You should describe better the growth conditions of seedlings (temperature, relative humidity, etc). Were the plants grown in a growth chamber? How many plants do you have per treatment?
You use only three replications for analysis. You must use more replications in future studies
RESULTS AND DISCUSSION
L76. Color alternatives. I think the word “alternatives” is not correct in that context.
Why do you discuss about the seed color? You should delete these results or justify better the importance of seed color in the study
Figure 2. The quality of this figure is poor and the error bars are not clearly visible. You should improve it
For the letters indicating the significance, consider using the always “a” for the biggest values and successive letters for lower values
The use of the word “assimilation” alone is not correct. Do you mean CO2 assimilation? Or you could write “photosynthesis performance” Or “Photosynthesis rate”
Figure 7D. You must add the error bars
L260-261: Add a reference to justify this
You must discuss more about the importance of the results and justify better if cotton could or not serve for Hg phytoremediation
CONCLUSION
L426-428. You should not include references in the conclusion. This sentence should be placed in the discussion section.
Round 2
Reviewer 1 Report
The manuscript has been revised and most of the reviewers' comments seem to have been taken into proper consideration.
There are still many mistakes, mainly related to lack of spaces e.g. between cottonseed, element symbol mercury should be written with capital letter, mistakes in spelling of units. Therefore, the manuscript should be re-read and these mistakes should be removed.
Reviewer 2 Report
Compared to the first version the work has improved, however, it is necessary to revise the English for the many errors present
Line 17-18: “ in response to some Hg concentrations” and insert the concentrations tested in brackets
Line 20: insert a space before over
Line 25: eliminate toxicity
Lines 60-65: I suppose two sentences should be eliminated
Line 69: revealed
Line 73: rephrase “as well as waving on biochemical substances”.
Line 90: the right word is grown?
Line 91: was tested
Line 92: variations
Lines 93-95: However, the germination percentage decreased starting at 1000 µM Hg, and dropped to 80%~90%. At 10000 µM Hg, the germination percentage decreased sharply to less than 10%.
Lines 109-110: The reduction in both shoots and roots in fresh and dry weight was dose-dependent (Table 1).
Lines 110-111: In shoots, there was no significant difference in fresh weight between control and 1 µM Hg
Lines 113-115: please rephrase the sentence
Line 120: there are two times µM. the same in lines 121- 123
Line 126: eliminate exogenous
Line 127: eliminate content
Line 129: eliminate “the ability of”
Line 132: use respectively instead of separately
Lines 207-208: rephrase the sentence in a clearer form
Line 211: eliminate “On the contrary”
Author Response
Please see the attachement.

Reviewer 3 Report
Thank you for the effort in correcting the errors, however there are still corrections to be made before it is accepted:
There are still many misspelled words. For example (misspelled word on the left; correct word on the right):
Champer – chamber (L369)
Grown – brown (L90)
Centmeter - Centimeter (L168)
Temperal – Temporal (L275)
Centration – concentration (L434)
Serve – severe (437)
Filed – Field (442)
Why do you use italics for some abbreviations? Follow the same criteria for all abbreviations
L20. Add a space between “inhibited” and “over”
L23. Why do you mention here the stem? You do not include data about stem biomass in the manuscript. You just show shoot and root biomass data in the manuscript.
L61-66. This part should be at the end of introduction and with the same formatting as the rest of the text. You must finish the final sentence: cotton are also….?
L81. “Currently” is not appropriate here. You could use “in the present study” or synonyms
Use superindex for μgg-1 in Table 2
L433-436. This sentence is very confusing and badly written. Please rewrite it to improve clarity
Author Response
Please see the attachement
